# Rapid, Cheap, and Effective COVID-19 Diagnostics for Africa

**DOI:** 10.3390/diagnostics11112105

**Published:** 2021-11-13

**Authors:** Lukman Yusuf, Mark Appeaning, Taiwo Gboluwaga Amole, Baba Maiyaki Musa, Hadiza Shehu Galadanci, Peter Kojo Quashie, Isah Abubakar Aliyu

**Affiliations:** 1Department of Medical Laboratory Science, College of Health Sciences, Bayero University Kano, Kano 700233, Nigeria; usuflukman41@yahoo.com; 2West African Centre for Cell Biology of Infectious Pathogens (WACCBIP), College of Basic and Applied Sciences, University of Ghana, P.O. Box LG54, Legon, Accra 23321, Ghana; mappeaning@st.ug.edu.gh; 3Department of Medical Laboratory Science, Faculty of Health and Allied Sciences, Koforidua Technical University, P.O. Box KF981, Koforidua 03420, Ghana; 4Africa Center of Excellence for Population Health and Policy, Bayero University Kano (ACEPHAP), Kano 700233, Nigeria; tayade10@yahoo.com (T.G.A.); babamaiyaki2000@yahoo.co.uk (B.M.M.); hgaladanci@yahoo.com (H.S.G.); 5Department of Community Medicine, Bayero University Kano, Aminu Kano Teaching Hospital, Kano 700233, Nigeria; 6Department of Medicine, College of Health Sciences, Bayero University Kano, Aminu Kano Teaching Hospital, Kano 700233, Nigeria; 7Department of Gynecology and Obstetrics, College of Health Sciences, Bayero University Kano, Kano 700233, Nigeria

**Keywords:** SARS-CoV-2, LAMP, recombinase amplification, COVID-19 diagnostics

## Abstract

Background: Although comprehensive public health measures such as mass quarantine have been taken internationally, this has generally been ineffective, leading to a high infection and mortality rate. Despite the fact that the COVID-19 pandemic has been downgraded to epidemic status in many countries, the real number of infections is unknown, particularly in low-income countries. However, precision shielding is used in COVID-19 management, and requires estimates of mass infection in key groups. As a result, rapid tests for the virus could be a useful screening tool for asymptomatic virus shedders who are about to come into contact with sensitive groups. In Africa and other low- and middle-income countries there is high rate of COVID-19 under-diagnosis, due to the high cost of molecular assays. Exploring alternate assays to the reverse transcriptase polymerase chain reaction (RT-PCR) for COVID-19 diagnosis is highly warranted. Aim: This review explored the feasibility of using alternate molecular, rapid antigen, and serological diagnostic assays to accurately and precisely diagnose COVID-19 in African populations, and to mitigate severe acute respiratory syndrome coronavirus 2 (SARS-CoV-2) RT-PCR diagnostic challenges in Africa. Method: We reviewed publications from internet sources and searched for appropriate documents available in English. This included Medline, Google Scholar, and Ajol. We included primary literature and some review articles that presented knowledge on the current trends on SARS-CoV-2 diagnostics in Africa and globally. Results: Based on our analysis, we highlight the utility of four different alternatives to RT-PCR. These include two isothermal nucleic acid amplification assays (loop-mediated isothermal amplification (LAMP) and recombinase polymerase amplification (RPA)), rapid antigen testing, and antibody testing for tackling difficulties posed by SARS-CoV-2 RT-PCR testing in Africa. Conclusion: The economic burden associated COVID-19 mass testing by RT-PCR will be difficult for low-income nations to meet. We provide evidence for the utility and deployment of these alternate testing methods in Africa and other LMICs.

## 1. Introduction

Severe Acute Respiratory Syndrome Coronavirus 2 (SARS-CoV-2) is highly infectious, akin to its predecessors SARS-CoV and Middle East respiratory syndrome coronavirus (MERS-CoV), which triggered epidemics in 2003 and an ongoing one since 2012, respectively [1]. The transmission of SARS-CoV-2 in Africa was initially slow, with few reported cases in Egypt following its index occurrence on 14 February 2020 [2]. In most African countries, many observers attributed the low recorded incidence rate of COVID-19 to under-diagnosis [2]. The disease had already spread rapidly across the world, with the World Health Organization (WHO) declaring Coronavirus disease 2019 (COVID-19) a pandemic on 24 February 2020. While globally, public health initiatives such as mass quarantine that have been instituted have been ineffective, the incidence of COVID-19 increased in the early days of the COVID pandemic with a high infection rate. The rapid spread of SARS-CoV-2 requires effective control in every part of the world with early case detection to halt transmission. This effective control of SARS-CoV-2, like other viruses, relies on finding and developing robust therapeutics, as well as simple, effective, and rapid diagnostics [3]. There is therefore an immediate global need to increase the diagnostic capacity everywhere. However, there is a global shortage of PCR reagents and swabs, as well as reports of discordant results from different COVID-19 tests. This could be due to the differences in the cycle threshold (Ct) cut-offs being used. Studies have analyzed the association between Cycle threshold (Ct) levels and the possibility of growing a live virus. It was reported that the Ct was much lower and log copies were significantly greater in those with live viral cultures, according to the findings. Other studies using Ct cut-off values ranging from Ct > 24 to Ct > 35 found no growth in the specimens. The likelihood of recovering the virus from specimens with Ct > 35 was calculated to be 8.3% (95% CI: 2.8% to 18.4%). Based on this, some kits/machines have lower cut-offs and other more stringent ones have higher Ct cut-offs. A way to address this is by repeated sampling over a few days—an expensive process by PCR. Therefore, the need for fast rapid diagnoses that aid in clinical decision-making and the need to consider alternate testing methods make the development of new options necessary. The aim of this review is to look into the possibility of using alternative molecular, rapid antigen, and serological diagnostic techniques to reliably and precisely detect COVID-19 in the African population, as well as to reduce SARS-CoV-2 RT-PCR diagnostic limitations.

## 2. Laboratory Diagnosis of SARS-CoV-2

During the early COVID-19 epidemic era, the availability of SARS-CoV-2 genome sequences, which were made available since 10 January 2020, facilitated the production of unique primers and standard laboratory protocols for the diagnosis of the virus [2]. The World Health Organization (WHO) subsequently recommended the use of quantitative reverse transcription polymerase chain reaction (qRT-PCR) for nucleic acid amplification as the gold standard for the diagnosis of active SARS-CoV-2 infection [4]. Under this recommendation, suspected shedders/cases, as well as asymptomatic and mildly symptomatic shedders/cases, were all to be confirmed by RT-PCR. Most current COVID-19 RT-qPCR-based tests target the ORF1ab region of the SARS-CoV-2 genome, in addition to the coding sequences of either the E or N proteins [5].

SARS-CoV-2 RNA can be identified from upper respiratory tract samples within 1–2 days before clinical symptoms occur. In mild cases, the persistence of the viral RNA has been recorded for 7–12 days. Virus shedding can last for up to 14 days in extreme cases. There have even been reports of sustained shedding of SARS-CoV-2 from nasopharyngeal fluids up to 24 days after symptom onset [2]. This does not, however, necessarily suggest patient’s infectiousness, possibly because of the non-viability of SARS-CoV-2 in such patients, as their specimens did not produce CPE on Vero E6 cell lines. The chances of false positive and negative results due to contamination, as well as bad and insufficient sampling, are an essential factor to consider during the selection of test methods. Nonetheless, although some molecular diagnostic kits for SARS-CoV-2 detection have been marketed, real-time RT PCR is still the most accepted choice of test.

## 3. Limitations of RT-PCR Testing

RT-PCR offers highly precise and sometimes quantitative SARS-CoV-2 RNA detection. It is, however, complicated, costly, and slow to execute. A single RT-PCR test kit can cost more than 100 US dollars, and it takes more than 15,000 US dollars to set up a diagnostic laboratory [6]. The analysis time of the RT-PCR available in most African countries is not less than 4 h, as depicted in Table 1, while the turn-around period is more than 24 h, from the sample collection to the readiness of the result. Although the Xpert Xpress SARS-CoV-2 test is a rapid automated in vitro diagnostic test that detects nucleic acid of SARS-CoV-2 with very short turn-around-time, it requires the use of expensive cartridges and/or machines, an example of which is the GeneXpert instrument system, and are expensive and require expertise.

In addition, molecular diagnostics are not accessible for many developing and underdeveloped countries due to their cost and the need for a trained clinical laboratory professionals and laboratories with a high complexity to operate. Furthermore, some studies have indicated a high discordant result rate for SARS-CoV-2 RT-PCR diagnostics, which may be due to inappropriate sample collection, purification, processing, and Ct, especially during the pandemic era [7]. Other factors can also cause false negative results, such as purified RNA degradation, the existence of RT-PCR inhibitors, or genomic mutations; some of these limitations have made the requirement for serology-based tests a necessity.

## 4. Constraints to COVID-19 Mass Testing by RT-PCR

### 4.1. Finance and Infrastructure

There are numerous advantages to massive population testing for COVID-19, as many high-income nations have shown. According to Liang et al. (2020), a higher COVID-19 mortality rate can be linked to fewer tests. However, most African countries are unable to achieve mass testing due to the shortage of large-scale laboratory testing capacities [2]. Most African countries faced many difficulties in their healthcare services prior to the advent of SARS-CoV-2, especially in terms of laboratory diagnostics, house-to-house case tracking, and community contact tracing for epidemiology. Nigeria, which is the most populous African country with over 206 million people, was only able to test 106,006 people across its 30 testing sites as of 19 June 2020, illustrating the lack of a laboratory testing capacity. This is because of the lack of test kits and qualified laboratory personnel as a result of the high demand for COVID-19 tests [2]. The acceptance and complexity of RT- PCR as the gold standard for COVID-19 diagnosis posed a significant testing impediment. This is due to the fact that this molecular assay’s equipment and kits are not cost effective, making them expensive to obtain in several African countries.

### 4.2. Turn-Around-Time for Report

Another significant limiting factor in PCR testing is the duration of the diagnosis, which is a limiting factor for achieving the required mass testing. Aside from the technological challenges of COVID-19 screening, seasonal variations could have an effect on the number of people tested. For example, enrolling people from rural villages and heavily populated shantytowns and communities would pose accessibility challenges during the rainy season [8,9]. In this circumstance, alternate quick and cheaper molecular tests and serological tests can be of immense help.

## 5. Alternate Molecular Testing Methods for COVID-19

Mass testing enables public health officials and other government policy makers to track the progress made in bringing this pandemic under control. The use of RT-PCR for mass testing not feasible in Sub-Saharan Africa and other low- and middle-income countries (LMICs) across the globe for reasons already enumerated above. An enormous number of COVID-19 RT-PCR tests needed to be conducted in order to contain the spread of the virus. This is difficult to do, even in high-income countries [10]. In addition, the early diagnosis of infection is important as it increases the likelihood of avoiding hospitalization and death. We present here two isothermal amplification techniques that have a parallel sensitivity and specificity, but lower cost and quicker reporting time compared with RT-PCR. These techniques can be harnessed by resource-limited countries to facilitate mass testing for COVID-19.

### 5.1. Loop-Mediated Isothermal Amplification

Loop-mediated isothermal amplification (LAMP) has some similarities to RT-PCR with respect to its results output, but the difference lies in its mode of amplification of DNA. LAMP amplifies DNA copies at a constant optimum temperature of about 65 °C, while RT-PCR involves a series of different temperatures and, as such, requires a thermo-cycler [11].

Briefly, LAMP involves the use of four different primers that specifically recognize six distinct regions (nucleocapsid protein gene region) on the target gene. The reaction occurs at a constant temperature using a strand displacement reaction. Amplification and detection of the genes occur in a single step where the mixture of samples, primers, DNA polymerase with strand displacement activity, and substrates are incubated at a constant temperature (about 65 °C). This provides a high amplification efficiency, with DNA being amplified by as much as 10^9^–10^10^ times within 15–60 min. Due its high specificity, the presence of an amplified product can indicate the presence of the target gene. In order for this technology to be applied in the diagnosis of RNA viruses, including SARS-CoV-2, a modification to include a reverse transcriptase enzyme is included in the process, in what is referred to as reverse transcription-loop-mediated isothermal amplification (RT-LAMP) (Figure 1). This enables the synthesis of complimentary DNA (cDNA) from RNA. Detection of the amplified product can be by visual turbidity, using the colorimetric method with the incorporation of SYBR Green I fluorescent dye, or by real time visualization using an inexpensive photometer [12].

RT-LAMP has been used to diagnose SARS-CoV-2 at a very high sensitivity and specificity compared with RT-PCR, which remains the gold standard as it is quick as 25 min [5]. In a study, statistical analyses using qRT-PCR as the gold standard indicated that the designed RT-LAMP assay had a sensitivity of 95.74% (95% Confidence interval: 89.97–100.00%), a specificity of 99.95% (95% Confidence interval: 99.86–100.00%), and a specificity of 99.95% (95% Confidence interval: 99.86–100.00%) [13].

Another study reported a sensitivity of 92.8% and specificity of 100% in identifying COVID-19 within the first nine days after the onset of infection [14]. RT-LAMP has also been optimized to identify SARS-CoV-2 from saliva specimen without the need for RNA extraction or purification [15]. The US Food and Drug Administration (FDA) has given Emergency Use Authorization for a number of RT-LAMP based point of care testing devices [16].

The main advantages of RT-LAMP for the diagnosis of SARS-CoV-2 is that it is fast and only requires a heating block. As such, it can be used at point-of care testing, for field studies, and in rural areas, unlike RT-PCR. Thus RT-LAMP should strongly be considered in African countries and other LMICs.

### 5.2. Recombinase Polymerase Amplification

Recombinase polymerase amplification (RPA) is an isothermal nucleic acid amplification technology that utilizes a recombinase enzyme to form complexes with oligonucleotide primers, and then pairs the primers with their complimentary sequences in double-stranded DNA. Single-stranded DNA binding (SSB) protein then binds to the displaced DNA strand and stabilizes it. Once a target sequence is present, the primer binds this sequence and DNA amplification then proceeds with the aid of polymerase. The amplification process continues rapidly, generating detectable levels of DNA within a few minutes from just a few target copies from the start of the amplification [17]. A reverse transcriptase can be added to the reaction mixture in order to be able to amplify RNA targets like SARS-CoV-2 and other RNA viruses. Amplification by RPA takes place at a much lower temperature than RT-PCR or RT-LAMP, optimally around 37–42 °C.

Different RPA-based assays have been developed to successfully detect different pathogens from different specimen [18,19,20,21]. In the wake of the COVID-19 pandemic, RT-RPA has been adapted and used to amplify the envelope protein (E) and RNA-dependent RNA polymerase (RdRP) genes of SARS-CoV-2 successfully, showing a good sensitivity and specificity compared with RT-PCR [22]. Again, RT-RPA has been used at point-of need for the detection of RdRP, E, and N genes of SARS-CoV-2. The RT-RPA achieved the following sensitivity and specificity: 94 and 100% for RdRP, 65 and 77% for E gene, and 83 and 94% for the N gene when compared with RT-PCR [23]. A much less expensive end point detection of the amplicon was demonstrated with the addition of SBYR Green I and with the use of a lateral flow strip [24].

Compared with other isothermal nucleic acid amplification techniques like the RT-LAMP, RT-RPA has a shorter run time (≤20 min for RT-RPA versus ≥30 min for RT-LAMP) and a lower amplification temperature (37 °C for RT-RPA versus 65 °C for RT-LAMP) [24]. Similar to RT-LAMP, RT-RPA is faster, cheaper, is forgiving of sample type, and is easier to deploy in LMICs than RT-PCR [25].

## 6. SARS-CoV-2 Antigen Testing as Alternative to RT-PCR Testing

One important diagnostic tool that can be harnessed to supplement the massive testing needed to curtail the spread of COVID-19 is the antigen-detecting rapid diagnostic test (Ag-RDTs). The development of the rapid antigen test for SARS-CoV-2 has been tremendous with the onset of the pandemic. As of 19 May 2021, the Foundation for Innovative New Diagnostics (FIND) listed 177 SARS-CoV-2 antigen-detection RDTs and 269 SARS-CoV-2 antibody detection RDTs that are currently marketed or in development [26].

Currently, most of the available Ag-RDTs are designed to directly detect nucleocapsid proteins produced by replicating virus in respiratory secretions [27]. They are therefore useful diagnostic tools like the RT-PCR tests for the diagnosis of active COVID-19 infection. The majority of SARS-CoV-2 Ag-RDTs that have received regulatory approval to date require nasopharyngeal samples [27]. A study showed that nasal swabs are equally, if not more, reliable than saliva swabs and can be used instead of nasopharyngeal swabs. In the future, they may prove to be a valuable diagnostic tool.

The accuracy of Ag-RDTs are affected by factors such as the duration of the infection, the viral load in the specimen, the quality and processing of the specimen collected from a person, and the precise formulation of the reagents in the test kits [28]. Nucleocapsid is the most abundant SARS-CoV-2 protein; this allows for increased sensitivity, but as nucleocapsids are highly conserved across coronaviruses, a false positive result may be reported as a result of the cross-reactivity between other human coronaviruses.

In the selection of Ag-RDTs for the above screening, the WHO recommends that selected kits meet the following ≥80% sensitivity and ≥97% specificity compared with the approved Nucleic Acid Amplification Test (NAAT) [29]. A comparative analysis of the performance of some four Ag-RDTs by different distributors in hospitalized COVID-19 patients showed that three out of the four kits were sensitive enough to identify symptomatic subjects infected with SARS-CoV-2 and with transmissible infection [30]. Another study found the Ag-RDTs by SD Biosensor, Biotical, and Panbioto be sensitive and specific enough to detect most of the contagious COVID-19 patients. Sensitivity in the ranges of 88.9% to 100% for samples with Ct < 26, with a specificity from 46.2% to 100%, were reported [31]. It is, however, recommended that Ag-RDT testing should only be conducted by trained professionals in order reduce the rate of false positive or false negative.

The main advantages of SARS-CoV-2 Ag-RDTs over RT-PCR are their speed, cost, and ability to be used as point of care (POC) devices and in the field, often generating results, in most cases, in ≤20 min. Unlike the alternate NAAT’s described here, most Ag-RDTs are not suitable and sensitive enough at very low viral loads and, as such, antigen RDT’s are not recommended for use in follow-up cases.

## 7. Serological Testing as Alternative to SARS-CoV-2 PCR Testing

Another major challenge with the control of COVID-19 is the incidence of asymptomatic infections and pre-symptomatic infections with elevated viral loads in the upper airways [32]. Widespread testing is therefore necessary to recognize asymptomatic, pre-symptomatic, and symptomatic infected individuals, and to allow for contact tracing and isolation. Although this has been extremely popular in countries such as Germany and South Korea, in most African countries where the infrastructure is poor, it is not usually feasible [33]. Tests are only possible for serious cases of suspected COVID-19 in many LMICs, and self-isolation is recommended for less severe cases [33]. Recorded cases and true incidence are also not equivalent. The infrastructure provided by academic and research laboratories and pharmaceutical companies is therefore being leveraged to further increase the testing capacity, as has been important in developed countries as well. The adoption of the sero-prevalence approach in Africa could be a beneficial way to increase the existing testing capacity, and could provide a quicker response time for high priority cases.

### Evaluation and Validation of Rapid Diagnostic Test

In order not to derail the progress made in controlling the spread of COVID-19, it is important to ensure that end users of various testing platforms have confidence in the quality and accuracy of results. The usefulness of serological tests in this pandemic is tightly hinged on the sensitivity and specificity of the assay [34]. To this end, many regulatory authorities across the globe have ensured stringent evaluation and validation of new manufacture serological assays for COVID-19. Although there are a lot of platforms for serological testing, including enzyme-linked immunosorbent assays (ELISAs), lateral-flow antibody assays (LFAs), bead-based assays based on Luminex technology, and automated serology platforms, most of the tests being evaluated across a number of African countries are based on LFAs [35,36]. This could be attributed to the fact that most countries in Sub-Saharan African have experience utilizing rapid diagnostic tests LFA devices for HIV and malaria, hence making LFAs the most preferred [37]. Some serological assays have shown promising outcomes for evaluation and validation. In one such report, a large number of positive results were observed in the cross-validation of 22 assays for the identification of IgM and IgG antibodies in patients with COVID-19 [34]; in this study, both ELISA and Lateral-flow tests were used [38].

The results in the test specificities of both antibody isotypes varied from 84% to 100%. One other lateral-flow assay was reported to have 100% specificity for both IgG and IgM [39].

## 8. The Value and Recommendations on the Usage of Serological Test

Serological testing should complement standard RT-PCR assays in symptomatic patients for the diagnosis of COVID-19. There is evidence that viral shedding in the upper respiratory system decreases dramatically 7 to 10 days after infection, resulting in poor swab results in at least 30 to 50% of cases of COVID-19 [40]. However, serological testing can also be used to assess SARS-CoV-2 infections retrospectively in individuals who have not previously tested positive with RT PCR [38]. The magnitude of the antibody response, on the other hand, is frequently linked to COVID-19 clinical severity [41]. Several serological assays have already been established, while both the viral spike and nucleocapsid’s binding antibodies indicates previous SARS-CoV-2 infection [42], the receptor-binding domain alone may be sufficient to detect antibody responses to SARS-CoV-2, it is also reported that its use may restrict cross-reactivity, which may arise from other coronavirus infections [43]. Due to the relative stability of the antibody response, serological tests are also recommended for mass screening of the population to determine the extent of the spread of the infection in the community. This is particularly important within the African context where testing with RT-PCR is very expensive and it is not practically possible to conduct mass testing. A survey conducted in Ghana with the use of rapid antibody test for COVID-19 reported higher community exposure than has officially been reported with RT-PCR [36,38]. Serological testing identifies individuals that have confirmed PCR positive-results, but have discordant serology tests. This identifies individuals whose immune systems require further study and opens up new fields of enquiry. Though not useful for diagnosis, rapid serological tests provide public health authorities with useful information on prevalence. This allows countries to plan and adapt their COVID-19 protocols. Serological approaches of testing have shown promising prospects in the fight against COVID-19.

## 9. Discussion

SARCoV-2 Ag-RDTs are indicated for use under some of the following circumstances: outbreak investigation or contact tracing where large numbers are going to be screened especially in very remote communities, monitoring trends in disease incidence especially among healthcare workers prior to contact with vulnerable populations and other frontline essential service providers, and where there is suspicion of community transmission [27]. In a bid to ease travel restrictions that were imposed by different government, Ag-RDT have been deployed to screen the travelling public at different ports of entry across different countries, with some success stories [28]. Indeed, airports such as Edmonton International Airport, have implemented some integrated COVID-19 testing that involves Ag-RDTs, RT-PCR and, in some cases, RT-LAMP.

Health authorities face many obstacles in the implementation of screening tests in low-resource environments, including the absence of facilities, trained staff, reagents, and state-of-the-art equipment, which hinders the broad testing and surveillance of COVID-19. The WHO promptly called for studies on point of care diagnostics to be used at the community level in this context. Despite the funds allocated by the Africa Centers for Disease Control and Prevention and the WHO for the procurement of diagnostic equipment and reagents, due to extreme shortages, these provisions have often not reached remote areas [9]. There is an ever-present need for the production of new tests with high internal quality management criteria and well-validated processes [44]. To provide solutions to the challenges faced in the molecular diagnosis of COVID-19, as well as improving the testing capacity, governments need to implement policies that will ensure proper distribution of SARS-CoV-2 rapid diagnostic kits.

## 10. Conclusions

Studies show that a majority of people in Sub-Saharan Africa have adequate knowledge of COVID-19, yet have a negative attitude towards COVID-19. Scaling up testing will provide evidence-based information to convince the African populace. However, this requires the precise implementation of high-volume diagnostics and quick use of the available data to help avoid further spread. It is clear that maximizing the number and availability of tests is of the utmost importance in combating the COVID-19 pandemic [44]. Research has shown that the frequency of testing and speed of diagnosis are the two most important factors for curbing COVID-19 incidence and the transmission of SARS-CoV-2. Thus, even low sensitivity tests are highly effective, if deployed at a high frequency [1,45].

We have highlighted several options for the rapid and cost-effective diagnosis of SARS-CoV-2 infection: RT-LAMP, RT-RPA, and Ag-RDTs. It is evident that the cost of these alternatives and the ability to scale up the number of tests make them more useful for population surveillance than RT-PCR for the diagnosis and monitoring of COVID-19 infection within LMICs. We thus recommend some of these approaches to be adopted by LMICs.

## Figures and Tables

**Figure 1 diagnostics-11-02105-f001:**
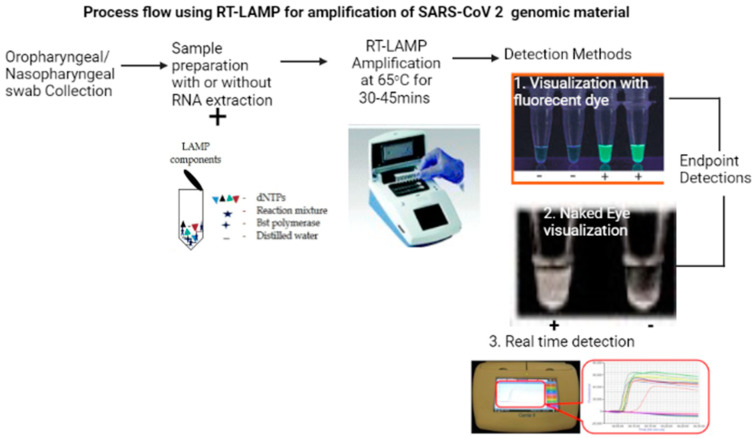
Using RT-LAMP for the diagnosis of COVID-19. Created with BioRender.com (accessed on 6 September 2021). Summary of procedures for RT-LAMP starting from swab collection, then RNA extraction and the addition of dNTPs, polymerase, distilled water, and reaction mix, which is then followed by amplification at 65 °C and, finally, detection by either the naked eye or florescent dye; fluorescent detection can be endpoint based or in real time.

**Table 1 diagnostics-11-02105-t001:** Some of the available SARS-CoV-2 diagnostic tests.

Test	Strength	Limitations	Time of Analysis	Cost (USD)
RT-PCR	High sensitivity and specificity	Requires expertise	4 h	300–6700
LAMP	High sensitivity and specificity, and a shorter turnaround time	Requires expertise	25 min	230–1550
Rcombinase Polymerase Amplification	High sensitivity and specificity, and a shorter turnaround time	Requires expertise	20 min	270
Antigen-RDT	Faster turnaround time and does not require expertise	Lower sensitivity	15 min	3–75
Antibody-RDT	Faster turnaround time and does not require expertise	Lower sensitivity	15 min	2.50–75

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
