# Peer review of "Rapid, Cheap, and Effective COVID-19 Diagnostics for Africa"

_diagnostics, 2021, doi:10.3390/diagnostics11112105_

Round 1
Reviewer 1 Report
Following the reviewers' suggestions, the authors have significantly improved their manuscript. Nevertheless, I suggest several modifications before endorsing the manuscript publication.
First of all, as indicated in the author's guidelines (https://www.mdpi.com/journal/diagnostics/instructions): Reviews: These provide concise and precise updates on the latest progress made in a given area of research. Systematic reviews should follow the PRISMA guidelines. Please, check and insert this missed part.
Moreover, in order to improve the manuscript, in the analysis of the strength and the limitation of each test for the SARS-CoV-2 identification, I suggest reading this missed reference: DOI= 10.3390/jcm9072026. Moreover, I suggesting inserting a new figure similar to figure 2 of the suggested reference, in order to produce a figure very useful for the readers.
Minor point:
Please, check all short forms: it should be mandatory to insert the extended form the first time and subsequently use everywhere the short form (for example, the authors have used WHO, World Health Organization indifferently).
Reviewer 2 Report
This well written review compares the efficacy of four alternatives to reverse-transcriptase polymerase chain reaction (RT-PCR) testing for SARS-CoV2 viral RNA: loop-mediated isothermal amplification (LAMP); recombinase polymerase amplification (RPA), and rapid antigen or antibody testing. The objective of the paper is to identify less expensive alternatives to RT-PCR testing which would be practical for middle and low income countries in Africa.
I am somewhat worried about whether there is an underlying assumption in this paper that mass testing and health passes will be accepted as necessary in Africa? Because there is a worldwide rejection of Gates’ “health passport” plan by the populations of most countries and in time politicians will realise this. Contrary to popular belief, COVID-19 is not a huge health problem, compared to the major killers of humans: heart disease, cancer, tuberculosis, malaria, malnutrition. COVID-19 mortality is on a par with influenza. In Western countries, the average age of COVID death is in the eighties, and is similar to the average age span.
It should be made clear if any of the authors have any conflicts of interest such as funding by the Gates Foundation or manufacturers of any of the tests cited in this paper. I’m sorry to say this, but the introduction is written in the style of a propaganda piece, rather than an objective scientific paper and I would like to see more evidence-based statements.
The references require improvement: please cite source articles rather than reviews and there are too many statements made without references to justify them (see list below).
Lines 68, 70, 74, 76, 92-93, 95, 102: give references please.
Lines 15 and 16: Obviously a scientific paper tends to look at what has happened in the past, but I believe it would be useful for you to get out ahead in your paper and present your conclusions in a way that addresses the future. The pandemic is already being downgraded to epidemic status in many countries, as scientific evidence accumulates that lockdowns, masks, and even vaccines don’t prevent virus transmission, and that this virus is here to stay, and like other coronaviruses, will be accepted as part of the human virome in the future. In addition, the delta variant is less pathogenic than other variants, therefore as with other new viruses, evolution favors less virulent pathogens. Therefore in future rapid tests for the virus may be a useful screen for asymptomatic virus shedders about to be in contact with vulnerable populations, such as carers of the elderly and sick, but it is becoming obvious that testing is not necessary for the population at large and that the push for mass testing is politically, not medically, driven.
I suggest you read and cite Ioannidis who showed that the majority of COVID deaths were in old age nursing homes.
Ioannidis JPA. Precision shielding for COVID-19: metrics of assessment and feasibility of deployment. BMJ Glob Health. 2021 Jan;6(1):e004614. doi: 10.1136/bmjgh-2020-004614.
I suggest you reframe your whole paper, taking out the justification for rapid tests as being for mass testing and exchanging it for more practical, real world, reasons: i.e.
- the early detection of infection because early intervention with treatment is more likely to prevent hospitalisation and mortality
- to prevent infection of the most vulnerable as outlined above
Lines 36 and 45: your statement is not true: SARS-CoV2 is not extremely virulent and has an over 99% survival rate. It is quite infectious though.
Line 44: substitute the word “ineffective” for “robust” because it is clear that mass quarantine – never before tried in a pandemic – is completely useless, and indeed it is becoming clear that lockdowns themselves cause greater mortality than the virus, due to missed hospital screenings, mental health consequences and an upsurge of suicides. The only actual benefits have been a possible downturn in road traffic accident fatalities.
Line 46: modify this sentence in light of what I wrote for Line 15.
Lines 49 and 60: references 3 and 2 are not really the best for these statements.
LIne 51 and Section 3 – RT-PCR limitations : it is the high rates of false positives, not false negatives, that are the predominant problem with SARS-CoV2 RT-PCR tests because of running the tests for too long and reporting high cycle thresholds (Ct) as positive instead of reporting the Ct and determining where Ct cut-off is in relation to ability to isolate the virus. This is something you should explain in your paper. Please read these papers:
Jefferson T, Spencer EA, Brassey J, Heneghan C. Viral cultures for COVID-19 infectious potential assessment - a systematic review. Clin Infect Dis. 2020 Dec 3:ciaa1764. doi: 10.1093/cid/ciaa1764.
Francis R, Le Bideau M, Jardot P, Grimaldier C, Raoult D, Bou Khalil JY, La Scola B. 2021. High-speed large-scale automated isolation of SARS-CoV-2 from clinical samples using miniaturized co-culture coupled to high-content screening. Clin Microbiol Infect. 27(1):128.e1-128.e7. doi: 10.1016/j.cmi.2020.09.018.
Line 64: do you mean suspected shedders or suspected cases or both? Please clarify. Are you aware that the word “cases” has been mis-used widely, especially in main stream media, for healthy RT-PCR positive people (who in fact in many instances are not even infected (i.e. false positives)? See paragraph above.
Line 87: why is the word cartridges in a different font?
Lines 95 and 96 are really saying the same thing: that there is a risk of false negative results due to a lack of a control for RNA integrity in RT-PCR tests – do none of the RT-PCR tests available in your country have an RNA control? This paragraph requires a section on false positives, which are the bigger problem, as described above.
Lines 101-102: Please supply evidence for this statement.
4.2 Turnaround time for reporting. You make a good practical point about the rainy season causing time delays in reporting, but lines 120-128 should be deleted as they are irrelevant to the topic of a virus test comparison.
Line 130 – delete – irrelevant to this section.
Lines 131-138 are more fitted to be in the introduction than here.
Line 144: which regions?
Line 158: RT-LAMP has been used to diagnose SARS-CoV-2 at very high sensitivity and specificity compare to RT-PCR which remains the gold standard as quick as 25 minutes (5).
This sentence doesn’t make sense. Are you saying that RT-LAMP can provide results as rapidly as within 25 minutes?
Section 5.1. This section requires the manufacturers of the RT-LAMPs whose sensitivity and specificity is being reported.
Where is Figure 1?
Was Figure 2 made by the authors? If not, please give the source. Where is the legend for this figure?
Line 184: a not are
Line 192: lower temperature than what? Please explain for readers who are not familiar with running PCR reactions.
The sentence in 199-201 would be better to follow that of Line 197, with the intervening sentence following.
Line 210: I’m sorry this sentence again sounds like propaganda rather than a scientific statement of fact. Where is your evidence that massive testing would curtail the spread of COVID?
Line 216: How do you know that nucleocapsid protein would only be in replicating virus and is not virus debris? Is there evidence for this statement?
Line 218: The majority …
I think that a brief note about nasal swabs vs saliva would be appropriate in this paragraph.
Line 222: Surely this statement applies to any kind of test?
Line 226: Please differentiate false positivity from cross-reactivity and explain that what you mean is that they may also pick up other human CoVs such as HCoV-229E (or whichever HCoVs you mean). In a false positive test there would be no CoV antigen present at all.
Line 231: Do you mean bid not bit? Could also say “In an attempt to …”
Line 234: Which airports? Be specific.
227-235 – isn’t this paragraph best placed in Discussion? And doesn’t it omit the most important possible use – screening by care workers prior to contact with vulnerable populations? I would also like to see a brief sentence or two about the possible disadvantage of mass screening – i.e. that it could theoretically interfere with the development of herd immunity, where the strongest members of society become naturally infected and immune and act as a firewall to protect the vulnerable.
Line 238: explain what NAAT stands for.
Lines 240, 279: space
Line 243: Explain Ct
Line 244: no such inference could be made!
Line 251: You MUST spell out abbreviations before using them. POC.
Line 288-290 is unnecessary.
Line 290: what about sensitivity? Duration of antibodies? Is there a test for cell mediated immunity?
Line 292 – why?
Line 297 – useful for long COVID.
Line 309. Serological testing also identifies individuals with confirmed PCR posi- 309
tive-results and discordant serology tests. What does this sentence mean? It doesn’t make sense.
Table 1 or a new table: I think readers will want a comparison of the sensitivity and specificity of these various techniques – would you be able to supply that information?
Is the cost of a single RT-PCR test correct in this table? Is it US dollars?
Surely there are a range of costs for the antigen and antibody tests?
Line 317 should not be 9. – it should be Discussion instead of this title.
Line 332: Again this is not true: look at the results in Uttar Pradesh where simple distribution of ivermectin has resulted in a far lower incidence of COVID death than, for example, the USA. I really think you need to look at the prevalence of COVID in various African countries and find out if there is a correlation between ivermectin use to control River Blindness, or hydroxychloroquine to control malaria and restructure your paper with a broader perspective, considering that while various tests have their place, there are other, often far cheaper, strategies which may be much more effective.
To summarise: the authors have a nice paper about test comparisons as they apply to Africa, but they have been somewhat tunnel-visioned in their approach, and I would like to see their work placed in the context of the whole picture concerning SARS-CoV2 and COVID 19.
Author Response
Please see the attachment. Using track changes

Round 2
Reviewer 1 Report
The authors have improved the manuscript following the reviewers' suggestions.
This manuscript is a resubmission of an earlier submission. The following is a list of the peer review reports and author responses from that submission.
Round 1
Reviewer 1 Report
This review aims to explore the feasibility of using alternate molecular, rapid antigen, and serological diagnostic assays to accurately and precisely diagnose covid-19 in the African population and to mitigate severe acute respiratory syndrome coronavirus 2 (SARS-CoV-2) RT-PCR diagnostic challenges in Africa. Based on their analysis, the authors suggest the utility of four different alternatives to RT-PCR.
At the end of the first section (introduction), the authors should insert the aims of the review.
Section 2 (Population immunity against SARS-CoV-2) could be deleted: it is unnecessary to the paper's aims.
Section 3 (Laboratory Diagnosis of SARS-CoV-2) should be revised. It is necessary to describe each laboratory test validated to SARS-CoV-2. Subsection 3.1 (LIMITATIONS OF RT-PCR TESTING) contains several wrong pieces of information.
For example, the authors stated that:
"The analysis time is not less than 4 hours, while the turnaround period is more than 24 hours from the sample collection to the readiness of result".
This is incorrect. For example, to date, there are several molecular tests that guarantee a fast diagnosis in less than an hour (i.e. Xpert® Xpress SARS-CoV-2).
If the purpose is to discuss the effectiveness of the diagnostic tests, the authors should perform an important literature review work inserting for each test strength and limitation, time of analysis, cost, and if it requires the presence of expert personnel, and other useful information (i.e. sample type) in order to achieve the aims of the study. in light of these important considerations, it should be important to insert a summary table. Moreover, I suggest revisiting all other sections.
Finally, it is important to include several missed references (i.e. Nawattanapaiboon et al . Colorimetric reverse transcription loop-mediated isothermal amplification (RT-LAMP) as a visual diagnostic platform for the detection of the emerging coronavirus SARS-CoV-2. Analyst. 2021 Jan 21;146(2):471-477. doi: 10.1039/d0an01775b. Epub 2020 Nov 9. PMID: 33165486.).
Reviewer 2 Report
Yusuf et al describe a comparative study to detect SARS-CoV2 aiming at using rapid and cheaper assays in LMICs such as RT-LAMP, RT-RPA or Ag-RDTs. This question is nevertheless of interest, since the early case detection is a need and all the tests compared to the gold standard RT-PCR technique have practical advantages. Unfortunately, there is no significant contribution to the field and a lot of papers talking about the same topic are avalaible.